# Trends in Antimicrobial Resistance of Canine Otitis Pathogens in the Iberian Peninsula (2010–2021)

**DOI:** 10.3390/antibiotics14040328

**Published:** 2025-03-21

**Authors:** Biel Garcias, Mar Batalla, Anna Vidal, Inma Durán, Laila Darwich

**Affiliations:** 1Department of Sanitat i Anatomia Animals, Veterinary Faculty, Universitat Autònoma de Barcelona, 08193 Cerdanyola del Vallès, Spain; biel.garcias@uab.cat (B.G.); mbatallafonoll@gmail.com (M.B.); anna.vidal@uab.cat (A.V.); 2Departamento de Veterinaria de Laboratorios Echevarne, 08037 Barcelona, Spain; idduran@laboratorioechevarne.com

**Keywords:** antimicrobial resistance, otitis, dogs, Iberian Peninsula

## Abstract

**Background**: The close relationship between humans and petsraises health concerns due to the potential transmission of antimicrobial-resistant (AMR) bacteria and genes. Bacterial otitis is an emerging health problem in dogs, given its widespread prevalence and impact on animal welfare. Early detection of resistance is vital in veterinary medicine to anticipate future treatment challenges. **Objective**: This study aimed to determine the prevalence of AMR bacteria involved in 12,498 cases of otitis in dogs from the Iberian Peninsula and the evolution of AMR patterns over an 11-year period. **Methods**: Data was provided by the Veterinary Medicine Department of a large private diagnostic laboratory in Barcelona. Antimicrobial susceptibility testing was performed using the standard disk diffusion method and minimum inhibitory concentration (MIC) testing. **Results**: The frequency of the principal bacterial agents was 35% *Staphylococcus* spp. (principally *S. pseudointermedius*), 20% *Pseudomonas* spp. (*P. aeruginosa*), 13% *Streptococcus* spp. (*S. canis*), and 11% Enterobacterales (*Escherichia coli* and *Proteus mirabilis*). Antimicrobial susceptibility testing revealed *P. aeruginosa* (among Gram-negatives) and *Enterococcus faecalis* (among Gram-positives) as the species with the highest AMR to multiple antimicrobial classes throughout the years. According to the frequency and time evolution of multidrug resistance (MDR), Gram-negative bacteria like *P. mirabilis* (33%) and *E. coli* (25%) presented higher MDR rates compared to Gram-positive strains like *Corynebacterium* (7%) and *Enterococcus* (5%). The AMR evolution also showed an increase in resistance patterns in *Proteus* spp. to doxycycline and *Streptococcus* spp. to amikacin. **Conclusions**: This information can be useful for clinicians, particularly in this region, to make rational antimicrobial use decisions, especially when empirical treatment is common in companion animal veterinary medicine. In summary, improving treatment guidelines is a key strategy for safeguarding both animal and human health, reinforcing the One Health approach.

## 1. Introduction

Antimicrobial resistance (AMR) has emerged as a global and critical challenge in the field of public health in recent decades [1,2]. The development and spread of antimicrobial resistance represent a growing and concerning threat to the efficacy of medical and veterinary treatments, thereby holding substantial implications for the health and well-being of both animals and humans [3]. With the growing close interaction between humans and companion animals [4] and the widespread use of broad-spectrum antimicrobial agents in veterinary medicine [5,6], assessing the potential risk of pet-to-human AMR transmission has become increasingly important. Despite this urgency, the data concerning the presence of AMR genes in companion animals remains limited [7], partly attributed to the challenges in systematically recording and monitoring clinical cases.

Canine otitis is a common condition seen in dogs at veterinary clinics worldwide, affecting 5% to 20% of the canine population [8,9,10,11]. Factors like ear anatomy, trauma with foreign bodies, moisture retention, endocrinopathies, autoimmune diseases, parasites, or allergies can predispose canines to opportunistic bacterial colonization [10,12,13]. These infections worsen symptoms and may lead to antibiotic resistance due to pathological changes such as hyperplasia, fibrosis, and biofilm formation. The issue of canine otitis emerges as a matter of significant importance due to its widespread prevalence and considerable impact on animal welfare.

Many Gram-positive and Gram-negative bacterial species have been linked to the pathogenesis of canine otitis, including *Staphylococcus* spp., *Pseudomonas* spp., *Proteus* spp., *Escherichia coli*, and *Corynebacterium* spp. [14,15,16,17]. Aminoglycosides and fluoroquinolones, like gentamicin and ciprofloxacin, are commonly used treatments, but rising resistance in zoonotic bacteria (notably *P. aeruginosa* and *S. pseudintermedius)* is a growing concern from a One Health perspective [14,18]. Therefore, a good knowledge of pathogen distribution and updated antimicrobial susceptibility data are essential for efficiently handling the condition.

While previous studies have delved into various aspects of canine otitis [18,19], a dedicated investigation within the Spanish context, focusing on the landscape of antimicrobial resistance in canine otitis, emerges as an imperative.The most recent studies on canine otitis have been conducted in Korea [14,20], Romania [16], and the Canarias Islands in Spain [21]. The Iberian Peninsula is a critical region for research due to its distinct climatic conditions, veterinary practices, and antimicrobial usage patterns, which may influence the development and spread of AMR in canine otitis. Furthermore, the region has a high incidence of otitis in dogs, yet comprehensive studies exploring the underlying resistance dynamics remain limited. Investigating AMR in this context is essential to inform regionally tailored treatment strategies, optimize antibiotic stewardship, and mitigate the potential transmission of resistant pathogens between animals and humans. Thus, previous studies conducted in the Iberian Peninsula have showna concerning prevalence of AMR in various canine pathologies [19,22], but none have focused on otitis over an extended period.

This study aimed to determine the most prevalent AMR bacteria involved in canine otitis in the Iberian Peninsula and to assess the patterns of evolution of antimicrobial resistance in dogs over an 11-year period (2010 to 2021), indentifying resistance trends and any emerging resistance patterns that may pose challenges to treatment options in the future. This research also seeks to provide essential insights that can guide clinicians in making informed decisions regarding empirical antimicrobial selection for treating infections in canine patients in the Iberian Peninsula, promoting an appropriate use of antimicrobials.

## 2. Results

### 2.1. Microbiological Diagnosis of Bacterial Infections

The analysis of canine otitis samples was conducted with 12,498 cases. The most important microbial genera are represented in Figure 1.

*Staphylococcus* spp. emerged as the most prevalent genus (34.7%), with *S. pseudintermedius* as the most relevant agent (Table 1). Other highly prevalent bacteria included *Pseudomonas* spp. (19.9%), mostly *P. aeruginosa*, *Streptococcus* spp. (13%), *Corynebacterium* spp. (8.9%), and *Enterococcus* spp. (6.6%). Among the *Streptococcus* spp. isolates, *S. canis* accounted for the majority (41%), while within the *Corynebacterium* genus *C. auriscanis* (52%) was the most prevalent species. *E. faecalis* (64%) emerged as the predominant species within the *Enterococcus* genus. Enterobacterales represented 11% of the cases, with *Escherichia coli* and *Proteus mirabilis* being the most important species (Table 1).

### 2.2. Antimicrobial Susceptibility Testing

A comprehensive assessment encompassed 8312 isolates, aiming to ascertain the antimicrobial susceptibility of Gram-positive and Gram-negative isolates. Specifically, the examined sample included 4566 *Staphylococcus* spp. isolates, 1711 *Streptococcus* spp. isolates, 1170 *Corynebacterium* spp. isolates, and 865 *Enterococcus* spp. isolates. For Gram-negative isolates, AST was conducted on 1439 retrieved Enterobacterales (comprising 754 *Escherichia* spp. and 685 *Proteus* spp. isolates) and 2620 *Pseudomonas* spp. isolates.

As regards the antimicrobial resistance profile, Gram-positive bacteria showed a more sensitive pattern than Gram-negative ones, presenting only notable resistance to aminopenicillins (>50%) and around 25–30% to amikacin, erythromycin, or clindamycin (Figure 2). By contrast, more than 50% of Gram-negative strains showed resistance to ampicillin, cephalosporines, trimethoprim-sulfamethoxazole, and chloramphenicol (Figure 2). According to the frequency of multidrug-resistance, 2% of Gram-positive strains presented an MDR pattern in comparison to the 10.9% of the Gram-negatives.

Among the Gram-positive group, *Enterococcus* spp. was the genus with the highest levels of AMR, with more than 70% of the isolates exhibiting resistance to 1st, 2nd, and 3rd generation cephalosporines, aminoglycosides, clindamycin, and Fusidic acid (Figure 3). Moreover, 5.1% of Enterococcus isolates showed a multidrug resistance pattern classified as MDR (3.7%) and extended-drug resistant (1.4% XDR). Other Gram-positive genera showing MDR patterns were Corynebacterium (7.4%) and Staphylococcus (1.1%).

On the other hand, Pseudomonas spp. exhibited the highest resistance rates among Gram-negative bacteria. The highest resistance rate was recorded for trimethoprim-sulfamethoxazole (>90%) and for amoxicillin-clavulanate, 1st, 2nd, and 3rd generation cephalosporines and chloramphenicol (>75%).

Within Enterobacterales, the AMR levels were relatively low, except for *Proteus* spp., where >60% of the isolates presented resistance to doxycycline (Figure 3). Moreover, 33% of Proteus strains were classified as MDR. The frequency of MDR strains was also high in *E. coli*, with 24.6% of MDR strains.

The evolution of MDR over the period of this study is shown in Figure 4, where the increasing percentage of MDR in Gram-negative strains, especially in *Proteus* spp. and *E. coli*, is evident. For Gram-positive strains, it is interesting to note the shift observed after 2017 in *Enterococcus* spp. and *Corynebacterium* spp. (Figure 4).

On the other hand, the evolution of AMR over time suggests an escalating resistance pattern for *Pseudomonas* in antimicrobial families like β-lactams, chloramphenicol, and trimethoprim + sulfametoxazole (Figure 5). Since 2016, *Enterococcus* has shown an increased resistance rate to several antimicrobial classes like β-lactams, aminoglycosides, clindamycin, and the Fusidic acid. A more restricted evolution was observed for *Staphylococcus* spp. and *Streptococcus* spp., which consistently showed a high resistance rate to ampicillin and amikacin, respectively. By contrast, both *Streptococcus* spp. and *S. pseudintermedius* have shown a progressive reduction in fluoroquinolone resistance since 2013. Finally, within Enterobacterales, from 2016 onward, *Proteus* spp. presented the highest resistance rates to doxycycline, reaching rates over 90% in 2019 (Figure 5).

Additionally, the comparison of AMR trends between the early (2010–2015) and late (2016–2021) study periods (Figure 6) showed similar patterns of evolution. Thus, *Pseudomonas* spp. registered the highest number of AMR, with a significant increase in resistance to almost all antimicrobial classes (β-lactams, macrolides, doxycycline, clindamycin, chloramphenicol), except for enrofloxacin that showed a reduction. *Enterococcus* spp. saw an increase in resistance to 1st and 3rd generation cephalosporines and clindamycin in the last period of study, but surprisingly showed increased sensitivity to aminopenicillins, doxycycline, fluoroquinolones, and chloramphenicol (Figure 6). An increase in resistance to aminopenicillins, 3rd generation cephalosporines, and clindamycin was also observed for *Corynebacterium* spp. in the last period. Finally, *Staphylococcus* spp. presented resistance to aminopenicillins, macrolides, aminoglycosides, and clindamycin, but the sensitivity pattern to doxycycline, fluoroquinolones, and Fusidic acid significantly increased., Similarly, *Streptococcus* spp. showed increased sensitivity to β-lactams, aminoglycosides, macrolides, fluoroquinolones, and chloramphenicol (Figure 6). As for Enterobacterales, the evolution of AMR in *E. coli* between both periods did not show any significant change, but for *Proteus* spp., the increased resistance levels to doxycycline, ciprofloxacin, clindamycin, Fusidic acid, and chloramphenicol were evident (Figure 6).

Alternatively, to analyze the correlations of AMR among different antimicrobial classes, correlograms were constructed separately for Gram-positive and Gram-negative bacteria (Figure 7). In general, the most common associations occurred within the same family, such as different generations of cephalosporins or enrofloxacin/ciprofloxacin. Notably, three clusters of co-resistance involving distinct antimicrobial classes were identified in Gram-negative bacteria: (1) chloramphenicol and trimethoprim-sulfamethoxazole alongside cephalosporins, (2) doxycycline with quinolones and aminoglycosides, and (3) amoxicillin and amoxicillin-clavulanic acid with clindamycin. In contrast, the associations in Gram-positive bacteria were less pronounced, though a relationship between cephalosporins and aminoglycosides was observed.

## 3. Discussion

Antimicrobial resistance (AMR) poses a significant threat to both human and animal health, requiring a comprehensive understanding of its evolution and impact. This study explored the evolution of antimicrobial resistance in canine otitis in the Iberian Peninsula, shedding light on the prevalent bacterial species, resistance patterns, and trends over an 11-year period. The findings of this study provide crucial insights that contribute to the broader understanding of AMR dynamics and inform better treatment practices.

The prevalence of bacterial genera in canine otitis samples reflects the complexity of this condition. *Staphylococcus* spp. emerged as the most prevalent genus, in line with previous studies highlighting its dominance [18,23]. *Pseudomonas* spp. and *Streptococcus* spp. also played substantial roles, consistent with global trends in otitis cases [18]. This distribution underscores the importance of accurate diagnosis and tailored treatment, as these bacteria exhibit varying resistance profiles.

*Pseudomonas aeruginosa* and *Staphylococcus pseudintermedius* emerged as the most encountered bacterial species. Similar trends have been reported in previous studies, validating the consistency of our findings [18,24,25].

The susceptibility of Enterobacterales to antimicrobial agents showed that *E. coli* displays lower AMR levels compared to *Proteus* spp., underscoring the divergent resistance dynamics within Gram-negative bacteria. The high rate of resistance of *E. coli* against ampicillin, likely influenced by its frequent use, raises concerns about the prudent use of antibiotics in pet clinics [26]. The stability of isolates over time is promising, though the rise in β-lactam resistance, particularly penicillins like ampicillin, emphasizes the need for cautious antibiotic administration. A significant shift from third-generation cephalosporins described in other studies, often implicated in resistance, adds further complexity to the landscape [27].

On the other hand, the antimicrobial resistance patterns of *Proteus* spp. showed the highest resistance rate to doxycycline (62.7%), with similar findings in analogous studies [26]. Notably, an evolutionary trend in resistance was discerned within *Proteus* spp., characterized by a significant increase in doxycycline resistance, followed by a notable decline in the final evaluation period of 2020–2021. Doxycycline is categorized as Class D by the European Medicines Agency (EMA), often indicating a preference for prioritization when selecting treatments in line with EMA guidelines. Based on our study findings, and aligning with the declining trend in resistance rates of doxycycline, more favorable alternatives could include aminoglycosides such as amikacin or gentamicin. Another viable option, classified under Category C, would involve the utilization of first- and second-generation cephalosporins.

Within the realm of antimicrobial resistance (AMR), the genus *Pseudomonas* spp. emerges as a noteworthy concern, exhibiting substantially heightened resistance compared to other Gram-negative bacterial genera, including Enterobacterales. In fact, an overarching trend of escalating resistance rates against various antimicrobial agents within the *Pseudomonas* genus from 2019 to 2021 is described in other studies [19,26,28]. Notably, this genus shares the highest levels of AMR with *Enterococcus* spp., highlighting the urgency of addressing its growing resistance burden in pets [19,28]. In clinical contexts, *Pseudomonas* spp. typically emerges not as primary pathogens but rather assuming a pivotal role in chronic otitis externa [29]. Delving into specific antimicrobial resistance profiles, high resistances were encountered, especially in β-lactams and cephalosporins. Interestingly, the trajectory of evolutionary resistance trends within the *Pseudomonas* genus aligns with the previously discussed observations, highlighting a persistent and escalating resistance pattern. Specifically, a significant subset of these bacterial isolates demonstrates resistance to β-lactams [30], emphasizing the gravity of this resistance mechanism.

*Pseudomonas aeruginosa* stands as a significant opportunistic pathogen implicated in the worldwide crisis of antibiotic resistance in the realm of human medical practice. Corresponding to a study carried out on rabbits within Spain [28], and in concordance with this study, it has been demonstrated that dealing with antimicrobial treatment for *P. aeruginosa* can be intricate. This complexity arises from a substantial proportion of the isolates exhibiting a resistant profile, encompassing antibiotics categorized as group B (third- and fourth-generation cephalosporins and fluoroquinolones), thereby curtailing the range of viable treatment options to carbapenems and polymyxins—both categorized as group A (antimicrobials, reserved for critical employment in human medical scenarios) [18]. Moreover, even aminoglycosides, belonging to group C, also constitute part of the restricted arsenal for potential use. Given this classification, aminoglycosides may emerge as the optimal choice for addressing pseudomonal infections.

Within the *Enterococcus* spp., notable patterns of antimicrobial resistance (AMR) emerge, revealing a substantial resistance landscape. The genus presents the highest resistance rates among all Gram-positive genera examined, especially against aminoglycosides and lincosamides. Insights from diverse studies conducted shed light on the intrinsic resistance characteristics of Enterococci, revealing susceptibility to cephalosporins, clindamycin, polymyxins, and aminoglycosides [19,31]. This intricately underscores the multifaceted resistance profile within the genus [21]. In contrast with our results, other studies described the highest prevalence of resistance within strains of *Enterococcus* spp. to tetracycline. Other studies align with the results obtained, highlighting the susceptibility of *Enterococcus* isolates to penicillin, ampicillin, and amoxicillin-clavulanate, and describing it as the most effective agents against enterococci [17,19,22]. In accordance with our findings and complementary research actions, while duly considering the classification stipulated by the European Medicines Agency (EMA), the optimal therapeutic avenues for combatting *Enterococcus* spp. infections manifest through the administration of ampicillin, penicillin, and amoxicillin-clavulanate. These agents demonstrate reduced resistance profiles and fall within the purview of categories D and C as designated by the EMA classification schema.

Examining *Staphylococcus* spp. isolates, the highest resistance rates were recorded for aminopenicillins. The evolutionary resistance trends within *Staphylococcus* spp. unveil significant dynamics, particularly evident in the case of ampicillin, where resistance escalated across evaluated time periods. In contrast, while penicillin exhibited substantial resistance rates, consistent with other periods studied [18], its resistance levels remained relatively stable over time. Interestingly, a parallel study notes a decline in fluoroquinolone resistance in both *P. aeruginosa* and *S. pseudintermedius* isolates since 2003, echoing findings within this research [19]. Furthermore, the reduction in fluoroquinolone resistance since 2013 in both *Streptococcus* and *S. pseudintermedius* isolates aligns with the implementation of the EcoAntibio plan, advocating for responsible antibiotic use [18].

Among the evaluated Gram-positive genera, *Streptococcus* spp. demonstrated the lowest resistance rates, underscoring its relatively favorable susceptibility profile. Nonetheless, the highest resistance rate within *Streptococcus* spp. isolates was noted for amikacin which presented an increased AMR levels during the examined period. This finding could threaten the effectiveness of aminoglycoside-based empirical treatments for streptococcal infections. Although there are only a few studies available, the research examining streptococcal resistance in canine otitis or in healthy dogs has offered valuable insights. These investigations have highlighted variations in resistance patterns, especially concerning enrofloxacin and gentamicin, which contrast with our findings [18]. But in general terms, the dynamic resistance evolution of *Streptococcus* spp. displayed generally low resistance percentages to various antimicrobial agents including aminopenicillins, amoxicillin-clavulanate, and trimethoprim/sulfamethoxazole, offering promising options for empirical therapy [19,26].

Certain limitations should be acknowledged in this study. First, data on age, sex, clinical history of the animal, and antimicrobial usage were not available; in consequence, these factors could not be assessed. Second, reliance on laboratory data could introduce a bias toward resistance, as some cases may have undergone empirical treatment before culture and susceptibility testing, and some cultures were more frequently requested for complicated cases than for uncomplicated ones.

Despite these limitations, the study of AMR correlations showed that AMR associations were most common within the same antibiotic family. However, in Gram-negative bacteria, three distinct co-resistance clusters were identified, involving cephalosporins, fluoroquinolones, aminoglycosides, and other classes. In Gram-positive bacteria, associations were weaker but included a link between cephalosporins and aminoglycosides. Overall, these findings suggest that horizontal gene transfer may contribute to the emergence of resistance. However, more specific studies are necessary to prove this hypothesis.

Horizontal gene transfer (HGT) is a key mechanism through which bacteria acquire and disseminate resistance genes. Resistance genes can be transferred through processes like conjugation, transformation, and transduction, and environmental conditions can either facilitate or hinder this transfer [32]. Moreover, it has been reported that antibiotic resistance genes can be shared among both commensal and MDR enteric bacteria in vitro by conjugative plasmid-mediated transfer [33]. The risk of HGT between pets and owners in the context of AMR is an important concern because it highlights how resistance genes can spread between humans and animals in a relatively close, shared environment. Further studies focused on HGT are a priority for developing global surveillance systems to track and monitor the spread of AMR genes in the environment, and to protect animal and public health.

## 4. Materials and Methods

### 4.1. Data Source and Management

A retrospective analysis of clinical microbiological data from the Veterinary Medicine Department of a private diagnostic laboratory located in Barcelona was conducted. The data analyzed included information on 13,141 bacterial isolates obtained from otic samples collected between January 2010 and January 2021 from companion dogs presented with suspected otitis. The laboratory records contained information about clinical cases submitted by veterinary clinics across Spain, Portugal, and Andorra.

The data was cleaned and assessed for duplicates and missing information, and only complete records were included in the analysis. The following variables of interest were extracted from the categorization of the records: sample origin, bacteria identification, antimicrobial susceptibility testing, and the geographic location of the case.

### 4.2. Microbiological Analysis and Antimicrobial Susceptibility Testing

Microbiological analysis and antimicrobial susceptibility testing were conducted by the laboratory following methodologies previously published by Li et al. [19] and Darwich et al. [22].

Microbial identification was accomplished through the utilization of the MALDITOF mass spectrometer or the API R ID system (bioMérieux, Madrid, Spain). Antimicrobial susceptibility testing adhered to the Performance Standards for Antimicrobial Susceptibility Testing for bacteria obtained from animals (M31-A3, CLSI VET01, 2008) and humans (M100-S24, CLSI, 2016) for drugs not approved for veterinary use (Appendix A). The laboratory has held the ISO-9001 quality management system certificate since 1998 and has been accredited by ENAC (National Accreditation Entity), in compliance with the criteria established in the ISO/IEC 17025 Standard, as outlined in the Technical Annexes 511/LE1947 for Pharmaceutical Toxicology and Microbiology.

Multidrug resistance (MDR) was defined according to Magiorakos et al. [34] as resistance to at least 1 agent in ≥3 antimicrobial categories, excluding intrinsic resistances from the analysis. In this study, an isolate was considered resistant to an antimicrobial category if it was classified as “non-susceptible to at least one agent in a category”. Extended drug resistance (XDR) was defined as resistance to 6 different antimicrobial families, while pan-drug resistance (PDR) was assigned to isolated with no available antibiotic treatment options.

### 4.3. Statistical Analysis

The data were imported into an accessible statistical software program, R Studio (version 4.3.1, The R Foundation for Statistical Computing) for the purpose of variable encoding and analysis. Both descriptive and statistical analyses were conducted. Graphical representations of antimicrobial resistance and MDR were generated using R version 4.2.0 [35], with the ggplot2 package [36] and ggcorrplot [37] as well as Microsoft Excel (version 2311, Microsoft 365 MSO).

## 5. Conclusions

In conclusion, the evolution of antimicrobial resistance in canine otitis in the Iberian Peninsula reflects a complex interplay of bacterial prevalence and resistance patterns. This study contributes valuable insights into the AMR landscape, highlighting trends, challenges, and opportunities for more targeted and informed treatment strategies. As AMR continues to threaten the efficacy of antimicrobial treatments, understanding its dynamics in veterinary settings becomes a crucial component of the broader effort to combat this global challenge.

## Figures and Tables

**Figure 1 antibiotics-14-00328-f001:**
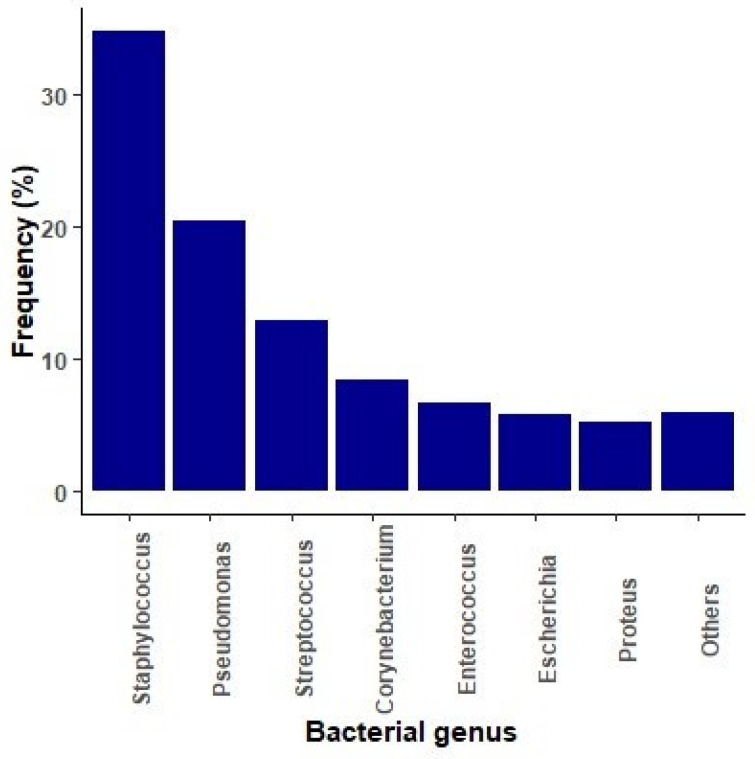
Bacterial genus distribution in otitis samples of dogs. The category “Others” includes: *Klebsiella* spp., *Enterobacter* spp., *Weisella* spp., *Acinetobacter* spp., *Serratia* spp., *Pantoea* spp., *Citrobacter* spp., *Bacillus* spp.

**Figure 2 antibiotics-14-00328-f002:**
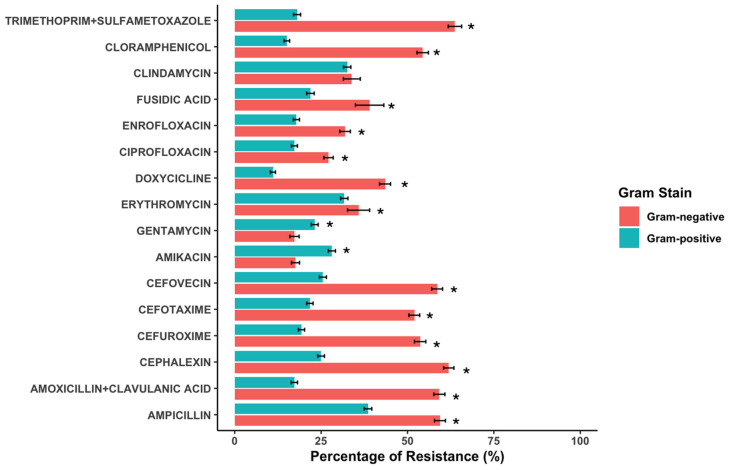
Comparison of antimicrobial resistance frequencies in Gram-negative (in red) and Gram-positive (in green) bacteria isolated from canine otitis. Confidence intervals (95%) are represented in bars. *p*-Values were analyzed using a chi-square test (* *p* < 0.05, statistically significance).

**Figure 3 antibiotics-14-00328-f003:**
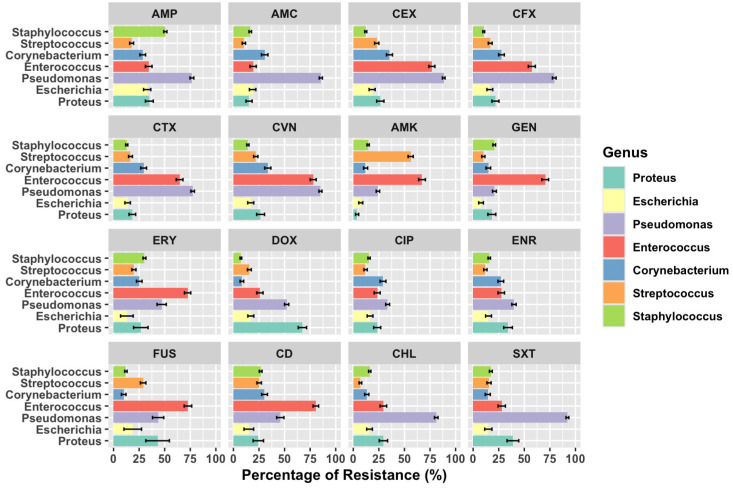
Comparison of AMR frequencies in bacterial isolates from canine otitis. Confidence intervals (95%) are represented in bars. Abbreviations: AMP (ampicillin), AMC (amoxicillin-clavulanate), CEX (cefalexin), CFX (cefuroxime), CTX (cefotaxime), CVN (cefovecin), AMK (amikacin), GEN (gentamicin), ERY (erythromycin), DOX (doxycycline), CIP (ciprofloxacin), ENR (enrofloxacin), FUS (Fusidic acid), CD (clindamycin), CHL (chloramphenicol), SXT (trimethoprim-sulfamethoxazole).

**Figure 4 antibiotics-14-00328-f004:**
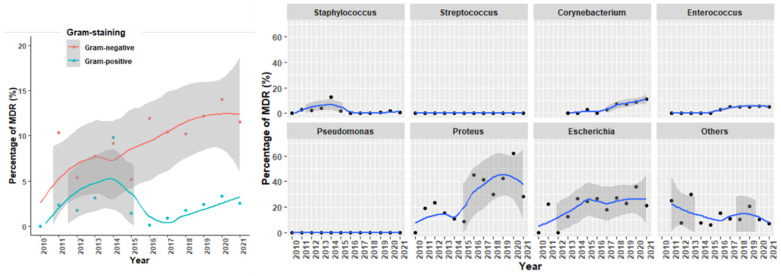
Evolution of MDR strains according to the gram-staining and the bacterial genus.

**Figure 5 antibiotics-14-00328-f005:**
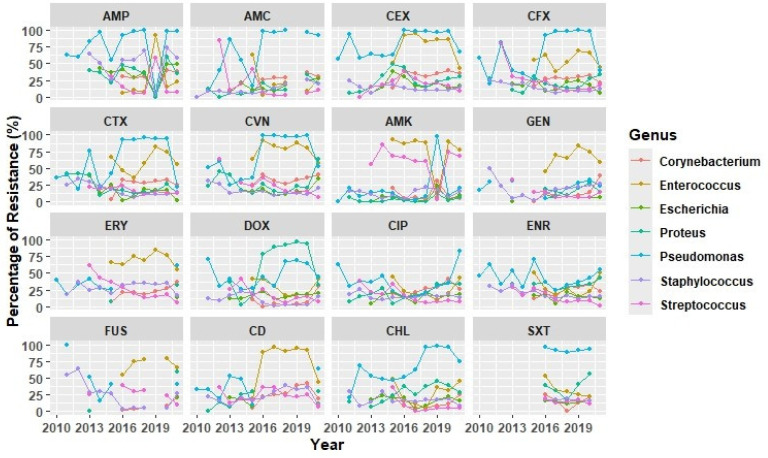
Line charts illustrate the fluctuation in overall resistance rates within tested otitis isolates encompassing distinct bacteria from 2010 and 2021. Abbreviations: AMP (ampicillin), AMC (amoxicillin-clavulanate), CEX (cefalexin), CFX (cefuroxime), CTX (cefotaxime), CVN (cefovecin), AMK (amikacin), GEN (gentamicin), ERY (erythromycin), DOX (doxycycline), CIP (ciprofloxacin), ENR (enrofloxacin), FUS (Fusidic acid), CD (clindamycin), CHL (chloramphenicol), SXT (trimethoprim-sulfamethoxazole).

**Figure 6 antibiotics-14-00328-f006:**
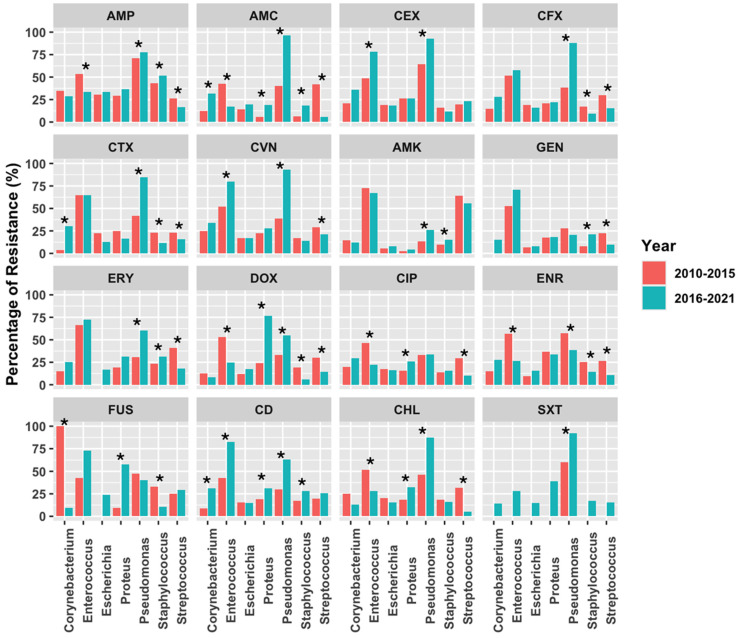
Comparison of AMR frequencies in bacterial isolates from canine otitis between two periods (2010–2015 and 2016–2021). Abbreviations: AMP (ampicillin), AMC (amoxicillin-clavulanate), CEX (cefalexin), CFX (cefuroxime), CTX (cefotaxime), CVN (cefovecin), AMK (amikacin), GEN (gentamicin), ERY (erythromycin), DOX (doxycycline), CIP (ciprofloxacin), ENR (enrofloxacin), FUS (Fusidic acid), CD (clindamycin), CHL (chloramphenicol), SXT (trimethoprim-sulfamethoxazole). *p*-values were analyzed using a chi-square test (* *p* < 0.05, statistically significance).

**Figure 7 antibiotics-14-00328-f007:**
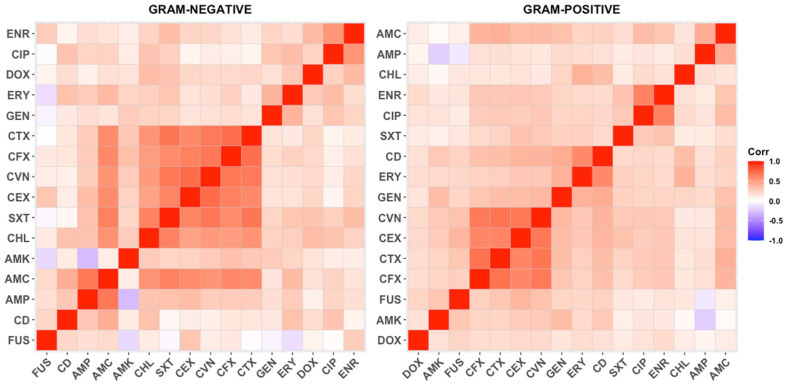
Correlation patterns of AMR for the different antibiotics and bacterial isolates. Axes are ordered using hierarchical clustering to show the most common associations. Abbreviations: AMP (ampicillin), AMC (amoxicillin-clavulanate), CEX (cefalexin), CFX (cefuroxime), CTX (cefotaxime), CVN (cefovecin), AMK (amikacin), GEN (gentamicin), ERY (erythromycin), DOX (doxycycline), CIP (ciprofloxacin), ENR (enrofloxacin), FUS (Fusidic acid), CD (clindamycin), CHL (chloramphenicol), SXT (trimethoprim-sulfamethoxazole).

**Table 1 antibiotics-14-00328-t001:** Number and frequencies of bacterial isolates identified in canine otitis samples. Only bacterial species belonging to the seven most common genomes with more than 10 samples are shown.

Microbiological Isolates	Number (% Within spp.)	Overall (N = 12,498) % (95% CI)
***Staphylococcus* spp.**	**n = 4339**	**34.7 (33.9–35.6)**
*S. pseudintermedius*	2024 (46.7)	16.2 (15.6–16.9)
*S.schleiferi*	451 (10.4)	3.6 (3.3–4)
*S. intermedius*	356 (8.2)	2.8 (2.6–3.2)
*S. aureus*	320 (7.4)	2.6 (2.3–2.9)
*S. epidermidis*	100 (2.3)	0.8 (0.7–1)
*S. simulans*	28 (0.6)	0.2 (0.2–0.3)
*S. chromogenes*	21 (0.5)	0.2 (0.1–0.3)
*S. warneri*	17 (0.4)	0.1 (0.08–0.2)
*S. hominis*	16 (0.4)	0.1 (0.07–0.2)
*S. haemolyticus*	14 (0.3)	0.1 (0.06–0.2)
Others	993 (22.9)	7.9 (7.5–8.4)
***Pseudomonas* spp.**	**n = 2541**	**20.3 (19.6–21.1)**
*P. aeruginosa*	2365 (93.1)	18.9 (18.2–19.6)
*P. putida*	25 (1.0)	0.2 (0.1–0.3)
*P. oryzihabitams*	15 (0.6)	0.1 (0.07–0.2)
Others	176 (6.9)	1.4 (1.2–1.6)
***Streptococcus* spp.**	**n = 1609**	**12.9 (12.3–13.5)**
*S. canis*	661 (41.1)	5.3 (4.9–5.7)
*S. halichoeri*	42 (2.6)	0.3 (0.2–0.5)
*S. dysgalacticae*	31 (1.9)	0.2 (0.2–0.4)
Others	875 (54.4)	7.0 (6.6–7.5)
***Corynebacterium* spp.**	**n = 1050**	**8.4 (7.9–8.9)**
*C. auriscanis*	545 (51.9)	4.4 (4–4.7)
*C. amycolatum*	201 (19.1)	1.6 (1.4–1.8)
Others	304 (29.0)	2.4 (2.2–2.7)
***Enterococcus* spp.**	**n = 842**	**6.7 (6.3–7.2)**
*E. faecalis*	538 (63.9)	4.3 (4–4.7)
*E. faecium*	66 (7.9)	0.5 (0.4–0.7)
*E. canintestini*	55 (6.5)	0.4 (0.3–0.6)
*E. avium*	26 (3.0)	0.2 (0.1–0.3)
*E. hirae*	10 (1.2)	0.1 (0.04–0.2)
Others	147 (17.5)	1.1 (1–1.4)
***Escherichia* spp.**	**n = 725**	**5.8 (5.4–6.2)**
*E. coli*	711 (98.1)	5.7 (5.3–6.1)
Others	14 (1.9)	0.1 (0.06–0.2)
***Proteus* spp.**	**n = 656**	**5.3 (4.9–5.7)**
*P. mirabilis*	622 (94.8)	5.0 (4.6–5.4)
*P. vulgaris*	18 (2.6)	0.1 (0.08–0.2)
Others	16 (2.4)	0.1 (0.08–0.2)

## Data Availability

Data are unavailable due to privacy or ethical restrictions.

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
