# Peer review of "Trends in Antimicrobial Resistance of Canine Otitis Pathogens in the Iberian Peninsula (2010–2021)"

_antibiotics, 2025, doi:10.3390/antibiotics14040328_

Round 1
Reviewer 1 Report
Comments and Suggestions for Authors
Overall Evaluation
The study is a valuable contribution to understanding AMR trends in canine otitis, particularly in Spain and Portugal. The manuscript is well-structured, though some sections need to be impr. Please see section-wise comments.
General comment:
This review report primarily focuses on the scientific aspects of the study. However, authors are encouraged to follow the nomenclature for bacterial species. For example, the phrase “Proteus spp.” Proteus is sometimes italicized and sometimes not. A similar pattern has been observed with other bacterial names. So correct it and be consistent throughout the manuscript.
Title
- Overall, the title accurately reflects the study’s focus on antimicrobial resistance (AMR) in dogs with otitis in the Iberian Peninsula over an 11-year period. However, the word “evolution” should be omitted because the study doesn’t involve any molecular, biological, or genomic adaptations of AMR. It solely focuses on the prevalence and temporal trends of AMR over the period of 11 years. I suggest that the title should be rephrased to something like this: temporal Trends in Antimicrobial Resistance of Canine Otitis Pathogens in the Iberian Peninsula (2010–2021). Having a clear title will enhance the readability and attract more genuine readers who are interested in this specific topic.
Abstract
The abstract presents a clear summary of the study, including key findings on bacterial prevalence and resistance trends. However, a few things that need to be addressed.
- The abstract lacks details on the methodology used for antimicrobial susceptibility testing (AST), which is critical for interpreting the results.
- The conclusion in the abstract could be more specific in terms of clinical implications: how should veterinarians adjust treatment based on these findings? OR how this research can be helpful in devising some preventive guidelines? Just need to add more specificity.
Introduction
The introduction effectively describes the importance of AMR in human and veterinary medicine. It discusses possible risk factors for AMR transmission, which is relevant and aligns with a One Health approach. However, here are a few points that can help improve the readability of this section.
- AMR is also a focus of this paper, but most importantly, it’s Canine Otitis. The disease’s pathology, epidemiology, and other related aspects should be addressed earlier in the introduction instead of in the fourth and second-to-last paragraphs. The early paragraphs discussing global AMR trends could be more concise but should include more of the latest and trending epidemiologic information on AMR in the context of Otitis.
- The study’s research gap is not explicitly stated—how does this research build upon previous studies conducted in Spain or internationally?
- Clarify how this study differs from prior research on AMR in canine otitis and justify why the Iberian Peninsula is a critical region for study.
Results
The study presents detailed tables and figures (e.g., Table 1, Figures 1–5) illustrating bacterial distribution and resistance trends in longitudinal analyses, identifying Pseudomonas aeruginosa and Enterococcus faecalis as the most clinically relevant resistant pathogens.
Here are a few areas to be improved in the result section.
- The figures lack statistical annotations. Figures 2 and 3 compare AMR frequencies, but no statistical significance markers (e.g., p-values and confidence intervals) are provided. I wonder why no statistical test has been performed. Without this information, it is difficult to determine whether observed differences are statistically meaningful or due to random variation.
- The study presents AMR trends from 2010 to 2021, but no clear comparison is made between early and late study periods. I wonder if grouping them into five-year intervals might be helpful to see how much has changed over time.
- Also, are there specific years when resistance patterns changed significantly? Consider summarizing key turning points in AMR trends that are shown in the figures but need to be explicitly stated in the text so readers can easily pick them up.
- The study states that 33% of Proteus spp. and 24.6% of E. coli were MDR, but no additional context is provided. Were certain antibiotic combinations more commonly resisted? Provide more context on MDR bacteria—what specific resistance patterns were most concerning?
Methods
- Overall, the study employs a well-structured methodology that includes clear explanations of microbiological identification techniques (MALDI-TOF and API systems), CLSI guidelines (M31-A3, VET01, M100-S24), as well as details on quality control and accreditation (ISO-9001 and ENAC). All of these methods and systems are well-established in diagnostics and certainly enhance the credibility of the study. Here are a few suggestions to improve this section further.
- The sample selection criteria could be more clearly defined. The study includes 12,498 otitis cases, but it is unclear whether any exclusion criteria were applied. For example, were dogs included regardless of prior antibiotic treatment? If so, how might this have influenced resistance patterns?
- No mention of whether breed, age, or other patient variables were considered, which could provide valuable insights into susceptibility differences.
One suggestion is to add a subsection explaining whether prior antibiotic use, breed, age, or clinical history were considered in the sample selection.
Next,
- The study states that AST followed CLSI guidelines, but specific breakpoints for resistance classification are not provided.
- How were multidrug-resistant (MDR), extensively drug-resistant (XDR), and pan-drug-resistant (PDR) strains defined? The study cites Magiorakos et al. (2012), but including an explicit definition in the methods section would improve clarity.
Even if you reference the studies in the manuscript, readers would appreciate having this fundamental information in this paper of interest as well. It would enhance the recognition of your work instead of leaving the reader uncertain and confused.
In my suggestion, it is important to clearly state the breakpoints used for resistance classification and to clarify the definitions of MDR, XDR, and PDR.
Finally,
- The study uses the Cochran-Armitage trend test, but no details are provided on how p-values or confidence intervals were determined.
- How were interannual variations in resistance trends statistically validated?
- No mention of potential confounders (e.g., changing antibiotic prescription patterns over time).
Suggested Revision: Provide more details on statistical methods, including significance thresholds and any adjustments for confounders.
Discussion:
A few suggestions to improve the discussion section.
- Any speculative wording should be removed if no scientific evidence exists. The third paragraph (lines 175-183) is about ampicillin resistance in E. coli. “The high rate of resistance of E. coli against ampicillin, likely influenced by its frequent use, raises concerns about the prudent use of antibiotics.” This statement was backed by another survey-based epidemiologic study, not any mechanistic biological study. It will be good to add some biological studies that support this evidence.
- I was very surprised that the horizontal gene transfer of AMR genes was not discussed throughout the paper, nor were any other environmental factors that are key drivers of AMR spread in those enlisted bacteria in this study. Here are possible papers.
- Thomas, Christopher M., and Kaare M. Nielsen. "Mechanisms of, and barriers to, horizontal gene transfer between bacteria." Nature reviews microbiology9 (2005): 711-721.
- Sher, A.A.; VanAllen, M.E.; Ahmed, H.; Whitehead-Tillery, C.; Rafique, S.; Bell, J.A.; Zhang, L.; Mansfield, L.S. Conjugative RP4 Plasmid-Mediated Transfer of Antibiotic Resistance Genes to Commensal and Multidrug-Resistant Enteric Bacteria In Vitro. Microorganisms 2023, 11, 193. https://doi.org/10.3390/microorganisms11010193
- The current discussion does not address limitations adequately. For example, this study only relies on the testing of AMR, not a mechanism of the AMR. No information on potential sampling bias, possible confounders, antibiotic usage trends in those patients, etc.
Author Response
Rebuttal letter
Thank you very much for taking the time to review this manuscript. The authors are vey grateful for your constructive revision of the manuscript. Your comments have been very useful to clearly improve the quality of this work.
Please find detailed responses below and the corresponding revisions/corrections highlighted/in track changes in the re-submitted files.
REVIEWER 1
The study is a valuable contribution to understanding AMR trends in canine otitis, particularly in Spain and Portugal. The manuscript is well-structured, though some sections need to be improved. Please see section-wise comments.
Comments 1: This review report primarily focuses on the scientific aspects of the study. However, authors are encouraged to follow the nomenclature for bacterial species. For example, the phrase “Proteus spp.” Proteus is sometimes italicized and sometimes not. A similar pattern has been observed with other bacterial names. So, correct it and be consistent throughout the manuscript.
Response 1: Thank you for pointing this out. We have reviewed and corrected the bacterial nomenclature throughout the manuscript.
Comments 2: Overall, the title accurately reflects the study’s focus on antimicrobial resistance (AMR) in dogs with otitis in the Iberian Peninsula over an 11-year period. However, the word “evolution” should be omitted because the study doesn’t involve any molecular, biological, or genomic adaptations of AMR. It solely focuses on the prevalence and temporal trends of AMR over the period of 11 years. I suggest that the title should be rephrased to something like this: temporal Trends in Antimicrobial Resistance of Canine Otitis Pathogens in the Iberian Peninsula (2010–2021). Having a clear title will enhance the readability and attract more genuine readers who are interested in this specific topic.
Response 2: We agree with this. The title of the revised version has been changed accordingly “Trends in Antimicrobial Resistance of Canine Otitis Pathogens in the Iberian Peninsula (2010–2021)”.
Comments 3: The abstract lacks details on the methodology used for antimicrobial susceptibility testing (AST), which is critical for interpreting the results.
Response 3: We have already included this information in the revised version (lines 17-20): “Data was provided by the Veterinary Medicine Department of a large private Laboratory of Diagnosis in Barcelona. The antimicrobial susceptibility testing was done using the standard disk diffusion method and the minimum inhibitory concentration testing”.
Comments 4: The conclusion in the abstract could be more specific in terms of clinical implications: how should veterinarians adjust treatment based on these findings? OR how this research can be helpful in devising some preventive guidelines? Just need to add more specificity.
Response 4: We have added more specific information regarding the clinical impact (lines 30-34): “This information can be useful for clinicians, particularly in this region, to make rational antimicrobial use decisions, especially when empirical treatment is common in companion animal veterinary medicine. In summary, improving treatment guidelines is a key strategy for safeguarding both animal and human health, reinforcing the One Health approach”.
Comments 5: Introduction. AMR is also a focus of this paper, but most importantly, it’s Canine Otitis. The disease’s pathology, epidemiology, and other related aspects should be addressed earlier in the introduction instead of in the fourth and second-to-last paragraphs. The early paragraphs discussing global AMR trends could be more concise but should include more of the latest and trending epidemiologic information on AMR in the context of Otitis.
Response 5: Agree. The introduction has been modified accordingly from line 50 to 79 and new references have been added accordingly: from refs 8 to 17, and refs 20 and 21.
Comments 6: The study’s research gap is not explicitly stated—how does this research build upon previous studies conducted in Spain or internationally?
Comments 7: Clarify how this study differs from prior research on AMR in canine otitis and justify why the Iberian Peninsula is a critical region for study.
Response 6 and 7: lines 68-79 have been modified accordingly.
Comments 8: The figures lack statistical annotations. Figures 2 and 3 compare AMR frequencies, but no statistical significance markers (e.g., p-values and confidence intervals) are provided. I wonder why no statistical test has been performed. Without this information, it is difficult to determine whether observed differences are statistically meaningful or due to random variation.
Response 8: Thank you for the comment. P-values were analyzed using chi-square test and all antibiotics with the exception of clindamycin presented significant differences (p<0.05) and this information has been indicated in the figure legend. We have also added confidence intervals to both figures (Fig 2 and 3).
Comments 9: The study presents AMR trends from 2010 to 2021, but no clear comparison is made between early and late study periods. I wonder if grouping them into five-year intervals might be helpful to see how much has changed over time.
Response 9: a new figure (Fig 6) has been added to the text comparing AMR trends between early and late study periods.
Comment 10: Also, are there specific years when resistance patterns changed significantly? Consider summarizing key turning points in AMR trends that are shown in the figures but need to be explicitly stated in the text so readers can easily pick them up.
Response 10: a new figure 6 has been added to the text comparing AMR trends between early and late study periods. Moreover, explicit information has been introduced in the text to facilitate the understanding of the readers (lines 187-203): “Alternatively, the comparison of AMR trends between early (2010-15) and late (2016-21) study periods (Figure 6), also showed similar patterns of evolution. Thus, Pseudomonas spp. registered the highest number of AMR, with a significant increase of resistance to almost all antimicrobial classes (β-lactams, macrolides, doxycycline, clindamycin, chloramphenicol), except for enrofloxacin that showed a reduction. Enterococcus spp. increased the resistance to 1G and 3G cephalosporines and clindamycin in the last period of study, but surprisingly increased the sensitivity to aminopenicillins, doxycycline, fluoroquinolones and chloramphenicol (figure 6). An increase of resistance to aminopenicillins and 3G cephalosporines and clindamycin was also seen for Corynebacterium spp. in the last period. Finally, Staphylococcus spp. presented resistance to aminopenicillins, macrolides, aminoglycosides and clindamycin, but the sensitivity pattern to doxycycline, fluoroquinolones and Fusidic acid increased significantly, as well as Streptococcus spp. increased sensitivity to β-lactams, aminoglycosides, macrolides, fluoroquinolones and chloramphenicol (Figure 6). As regards Enterobacterales, the evolution of AMR for E. coli between both periods did not show any significant change, however for Proteus spp. the increased resistance levels to doxycycline, ciprofloxacin, clindamycin, fusidic acid and chloramphenicol was evident (Figure 6).”
Comments 11: The study states that 33% of Proteus spp. and 24.6% of E. coli were MDR, but no additional context is provided. Were certain antibiotic combinations more commonly resisted? Provide more context on MDR bacteria—what specific resistance patterns were most concerning?
Response 11: We have added a new figure 7 showing the correlation of AMR patterns among antibiotics and Gram-negative and positive bacteria. The new text is found in the results (lines 212-221): “Alternatively, to analyze the correlations of AMR among different antimicrobial classes, correlograms were constructed separately for Gram-positive and Gram-negative bacteria (Figure 7). In general, the most common associations occur within the same family, such as different generations of cephalosporins or enrofloxacin/ciprofloxacin. Notably, three clusters of co-resistance involving distinct antimicrobial classes were identified in Gram-negative bacteria: (1) chloramphenicol and trimethoprim-sulfamethoxazole alongside cephalosporins, (2) doxycycline with quinolones and aminoglycosides, and (3) amoxicillin and amoxicillin-clavulanic acid with clindamycin. In contrast, the associations in Gram-positive bacteria were less pronounced, though a relationship between cephalosporins and aminoglycosides was observed.”
And in the discussion in lines 346-356: “Despite these limitations, the study of AMR correlations showed that AMR associations were most common within the same antibiotic family. However, in Gram-negative bacteria, three distinct co-resistance clusters were identified, involving cephalosporins, fluoroquinolones, aminoglycosides, and other classes. In Gram-positive bacteria, associations were weaker but included a link between cephalosporins and aminoglycosides. Overall, these findings suggest that horizontal gene transfer may contribute to the emergence of resistance. However, more specific studies are necessary to prove this hypothesis.”
Comments 12: The sample selection criteria could be more clearly defined. The study includes 12,498 otitis cases, but it is unclear whether any exclusion criteria were applied. For example, were dogs included regardless of prior antibiotic treatment? If so, how might this have influenced resistance patterns? No mention of whether breed, age, or other patient variables were considered, which could provide valuable insights into susceptibility differences. One suggestion is to add a subsection explaining whether prior antibiotic use, breed, age, or clinical history were considered in the sample selection.
Response 12: the inclusion criteria applied was already stated in the text (The data was cleaned and assessed for duplicates and missing information, and only complete records were included in the analysis. The following variables of interest were extracted from the categorization of the records: sample origin, bacteria identification, antimicrobial susceptibility testing and the geographic location of the case (lines 338-341). Moreover, all animals were included regardless of prior antibiotic treatment, since no information was recorded about this usage. Also, information of breed, sex or age was not available. For this reason, we have included a paragraph of the study limitations (lines 333-338): “Certain limitations should be acknowledged in this study. First, data on age, sex or clinical history of the animal and antimicrobial usage were not available, in consequence, these factors could not be assessed. Second, relying on laboratory data could introduce a bias toward resistance, as some cases may have undergone empirical treatment before culture and susceptibility testing, and some cultures were more frequently requested for complicated cases than for uncomplicated ones”.
Comments 13: The study states that AST followed CLSI guidelines, but specific breakpoints for resistance classification are not provided.
Response 13: Specific breakpoints reported by the Echevarne Lab can be found as annex 1 document.
Comments 14: How were multidrug-resistant (MDR), extensively drug-resistant (XDR), and pan-drug-resistant (PDR) strains defined? The study cites Magiorakos et al. (2012), but including an explicit definition in the methods section would improve clarity.
Response 14: Thanks for the suggestion. We have added the definition of each category in the text as follows (lines 391-395): “In the definitions proposed for MDR in this study, an isolate is considered resistant to an antimicrobial category when it is ‘non-susceptible to at least one agent in a category’. Extended drug resistance (XDR) was considered when isolates were resistant to 6 different families and Pan drug resistance (PDR) when no options existed for antibiotic treatment.”
Comments 15: The study uses the Cochran-Armitage trend test, but no details are provided on how p-values or confidence intervals were determined. No mention of potential confounders (e.g., changing antibiotic prescription patterns over time). Suggested Revision: Provide more details on statistical methods, including significance thresholds and any adjustments for confounders.
Response 15: Cochran-Armitage trend test was not applied in the analysis, so we have removed this from the text. We have provided more statistical information about correlation analyses, significant differences (Chi-Square tests) and added 95%CI in the results. Unfortunately, we could not introduce the confounders in the analysis since this information was not recorded in the database.
Comments 16: Any speculative wording should be removed if no scientific evidence exists. The third paragraph (lines 175-183) is about ampicillin resistance in E. coli. “The high rate of resistance of E. coli against ampicillin, likely influenced by its frequent use, raises concerns about the prudent use of antibiotics.” This statement was backed by another survey-based epidemiologic study, not any mechanistic biological study. It will be good to add some biological studies that support this evidence.
Response 16: We have included a more appropriate reference, ref 27: Reynaert WC. An overview of the antimicrobial resistance mechanisms of bacteria. AIMS Microbiol. 2018 Jun 26;4(3):482-501. doi: 10.3934/microbiol.2018.3.482.
Comments 17: I was very surprised that the horizontal gene transfer of AMR genes was not discussed throughout the paper, nor were any other environmental factors that are key drivers of AMR spread in those enlisted bacteria in this study. Here are possible papers.
Thomas, Christopher M., and Kaare M. Nielsen. "Mechanisms of, and barriers to, horizontal gene transfer between bacteria." Nature reviews microbiology9 (2005): 711-721.
Sher, A.A.; VanAllen, M.E.; Ahmed, H.; Whitehead-Tillery, C.; Rafique, S.; Bell, J.A.; Zhang, L.; Mansfield, L.S. Conjugative RP4 Plasmid-Mediated Transfer of Antibiotic Resistance Genes to Commensal and Multidrug-Resistant Enteric Bacteria In Vitro. Microorganisms 2023, 11, 193. https://doi.org/10.3390/microorganisms11010193
Response 17: a new section in the discussion has been added to reflect the relevance of horizontal gene transfer of AMR (lines 346-356):
“Horizontal gene transfer (HGT) is a key mechanism through which bacteria acquire and disseminate resistance genes. Resistance genes can be transferred through processes like conjugation, transformation, and transduction, and environmental conditions can either facilitate or hinder this transfer [32]. Moreover, it has been reported that antibiotic re-sistance genes can be shared among both commensal and MDR enteric bacteria in vitro by conjugative plasmid-mediated transfer [33]. The risk of HGT between pets and owners in the context of AMR is an important concern because it highlights how resistance genes can spread between humans and animals in a relatively close, shared environment. Fur-ther studies focused on HGT are a priority for developing global surveillance systems to track and monitor the spread of AMR genes in the environment, and to protect animal and public health”.
Comments 18: The current discussion does not address limitations adequately. For example, this study only relies on the testing of AMR, not a mechanism of the AMR. No information on potential sampling bias, possible confounders, antibiotic usage trends in those patients, etc.
Response 18: we have included a paragraph of the study limitations (lines 333-338): “Certain limitations should be acknowledged in this study. First, data on age, sex or clinical history of the animal and antimicrobial usage were not available, in consequence, these factors could not be assessed. Second, relying on laboratory data could introduce a bias toward resistance, as some cases may have undergone empirical treatment before culture and susceptibility testing, and some cultures were more frequently requested for complicated cases than for uncomplicated ones”.
Reviewer 2 Report
Comments and Suggestions for Authors
Materials and methods
Data sources:
The information regarding breed/age/season specific data could be incorporated if it is available
Year wise resistant pattern of each antibiotic could be incorporated as a separate paragraph in the text will helps to understand the evolution of AMR rather giving in the form of chat alone and discussion must be incorporated in a separate paragraph pertaining to this
Line no-175-E.coli should be in Italics in all the text
The significance of this findings (novelty) should be emphasized in public health context
May be considered for short communication rather full length paper
Author Response
Rebuttal letter
Thank you very much for taking the time to review this manuscript. The authors are very grateful for your constructive revision of the manuscript. Your comments have been very useful to clearly improve the quality of this work. Please find detailed responses below and the corresponding revisions/corrections highlighted/in track changes in the re-submitted files.
REVIEWER 2
Comment 1: Data sources:The information regarding breed/age/season specific data could be incorporated if it is available.
Response 1: we totally agree with the comment, however, unfortunately this information was not recorded in the database, so we could not introduce them in the analyses. We have described this issue as a limitation factor of this study.
(lines 333-338): “Certain limitations should be acknowledged in this study. First, data on age, sex or clinical history of the animal and antimicrobial usage were not available, in consequence, these factors could not be assessed. Second, relying on laboratory data could introduce a bias toward resistance, as some cases may have undergone empirical treatment before culture and susceptibility testing, and some cultures were more frequently requested for complicated cases than for uncomplicated ones”.
Comment 2: Year wise resistant pattern of each antibiotic could be incorporated as a separate paragraph in the text will helps to understand the evolution of AMR rather giving in the form of chat alone and discussion must be incorporated in a separate paragraph pertaining to this.
Response 2: a new figure 6 has been added to clarify these results.
Comment 3: Line no-175-E.coli should be in Italics in all the text
Response 3: done it.
Comment 4: The significance of this findings (novelty) should be emphasized in public health context.
Response 4: A new paragraph has been included in the abstract emphasizing the novelty of this study: “This information can be useful for clinicians, particularly in this region, to make rational antimicrobial use decisions, especially when empirical treatment is common in companion animal veterinary medicine. In summary, improving treatment guidelines is a key strategy for safeguarding both animal and human health, reinforcing the One Health approach.”
In addition, in the discussion the following section (lines 346-356) was included reinforcing the role of AMR gene transfer among pets and humans.
Comment 5: May be considered for short communication rather full length paper
Response 5: since another reviewer asked for more information and figures, it will be difficult to reduce all this data as a short paper. However, if the editor considers it necessary, we will try to reduce the contents.
Reviewer 3 Report
Comments and Suggestions for Authors
Comments and Suggestions for the Authors
The ‘article’ manuscript “Prevalence and Evolution of Antimicrobial Resistance Bacteria isolated from Otitis in Dogs of the Iberian Peninsula from 2010 to 2021.” gives an overview of antimicrobial resistant bacteria prevalent in Dog Otitis in Iberian Peninsula. I recommend the manuscript for Minor modifications/rectifications before publication. I have a few minor comments and suggestions to further enhance the manuscript.
- Title reflects the main subject of the manuscript.
- Abstract is good, that summarizes and reflects the topic described in the manuscript.
- Key words are adequate.
- The introduction is adequate.
- All figures need to be presented with better resolution images.
- The article lacks consistency in terminology (always italicizing bacterial genus/species names; make changes in line number 175, 176, 177, 197, 198, 200, 225, 234 and 248).
- Authors need to explain resistance trends in clear two different sub sections for Gram-positive and Gram-negative bacteria. The current presentation is okay, but sub sections would be better for understanding.
- References are adequate.
Current Quality of English is okay.
Author Response
Rebuttal letter
Thank you very much for taking the time to review this manuscript. The authors are very grateful for your constructive revision of the manuscript. Your comments have been very useful to clearly improve the quality of this work.
Please find detailed responses below and the corresponding revisions/corrections highlighted/in track changes in the re-submitted files.
REVIEWER 3
I recommend the manuscript for Minor modifications/rectifications before publication. I have a few minor comments and suggestions to further enhance the manuscript.
Title reflects the main subject of the manuscript.
Abstract is good, that summarizes and reflects the topic described in the manuscript.
Key words are adequate.
The introduction is adequate.
All figures need to be presented with better resolution images.
The article lacks consistency in terminology (always italicizing bacterial genus/species names; make changes in line number 175, 176, 177, 197, 198, 200, 225, 234 and 248).
Response: thanks for the suggestion. We have reviewed and corrected all names.
Authors need to explain resistance trends in clear two different sub sections for Gram-positive and Gram-negative bacteria. The current presentation is okay, but sub sections would be better for understanding.
Response: the manuscript has suffered more improvements and added new figures 6 and 7, showing correlations among antibiotic classes between Gram-positive and negatives that we think are improving the understanding of this paper.
References are adequate.
Response: thank you for your comments.
Round 2
Reviewer 2 Report
Comments and Suggestions for Authors
The authors have significantly revised the manuscript as per reviewer's comments. Hence accepted.